# Relationship between Circulating Lipids and Cytokines in Metastatic Castration-Resistant Prostate Cancer

**DOI:** 10.3390/cancers13194964

**Published:** 2021-10-01

**Authors:** Hui-Ming Lin, Nicole Yeung, Jordan F. Hastings, David R. Croucher, Kevin Huynh, Thomas G. Meikle, Natalie A. Mellett, Edmond M. Kwan, Ian D. Davis, Ben Tran, Kate L. Mahon, Alison Zhang, Martin R. Stockler, Karen Briscoe, Gavin Marx, Patricia Bastick, Megan L. Crumbaker, Anthony M. Joshua, Arun A. Azad, Peter J. Meikle, Lisa G. Horvath

**Affiliations:** 1Garvan Institute of Medical Research, Darlinghurst, NSW 2010, Australia; h.lin@garvan.org.au (H.-M.L.); n.yeung@garvan.org.au (N.Y.); j.hastings@garvan.org.au (J.F.H.); d.croucher@garvan.org.au (D.R.C.); Kate.Mahon@lh.org.au (K.L.M.); m.crumbaker@garvan.org.au (M.L.C.); anthony.joshua@svha.org.au (A.M.J.); 2St Vincent’s Clinical School, UNSW Sydney (the University of New South Wales), Darlinghurst, NSW 2010, Australia; 3The Kinghorn Cancer Centre, St Vincent’s Hospital, Darlinghurst, NSW 2010, Australia; 4Metabolomics Laboratory, Baker Heart and Diabetes Institute, Melbourne, VIC 3004, Australia; Kevin.Huynh@baker.edu.au (K.H.); Thomas.Meikle@baker.edu.au (T.G.M.); Natalie.Mellett@baker.edu.au (N.A.M.); peter.meikle@baker.edu.au (P.J.M.); 5Department of Medical Oncology, Monash Health, Clayton, VIC 3168, Australia; edmond.kwan@monash.edu; 6Department of Medicine, School of Clinical Sciences, Monash University, Clayton, VIC 3168, Australia; 7Eastern Health Clinical School, Monash University, Box Hill, VIC 3128, Australia; Ian.Davis@monash.edu; 8Cancer Services, Eastern Health, Box Hill, VIC 3128, Australia; 9Sir Peter MacCallum Department of Oncology, University of Melbourne, Parkville, VIC 3010, Australia; Ben.Tran@petermac.org (B.T.); arun.azad@petermac.org (A.A.A.); 10Department of Medical Oncology, Peter MacCallum Cancer Centre, Melbourne, VIC 3000, Australia; 11Medical Oncology, Chris O’Brien Lifehouse, Camperdown, NSW 2050, Australia; alisonwestmead@gmail.com (A.Z.); martin.stockler@sydney.edu.au (M.R.S.); 12Royal Prince Alfred Hospital, Camperdown, NSW 2050, Australia; 13Faculty of Medicine and Health, University of Sydney, Camperdown, NSW 2050, Australia; 14Concord Cancer Centre, Concord Repatriation General Hospital, Concord, NSW 2139, Australia; 15Mid North Coast Cancer Institute, Coffs Harbour, NSW 2450, Australia; karen.briscoe@ncahs.health.nsw.gov.au; 16SAN Integrated Cancer Centre, Sydney Adventist Hospital, Wahroonga, NSW 2076, Australia; gmarx@nhog.com.au; 17Medical Oncology, St George Hospital, Kogarah, NSW 2217, Australia; Patricia.Bastick@health.nsw.gov.au

**Keywords:** circulating lipids, cytokines, metastatic castration-resistant prostate cancer, lipidomic, biomarkers

## Abstract

**Simple Summary:**

Lipids (fatty substances) and cytokines are molecules that affect how the immune response works. The measurement of the amounts of lipids and cytokines in blood might give clues about how prostate cancers grow or respond to treatment. This study looked at the blood levels of lipids and cytokines in men with advanced prostate cancer that was growing despite standard treatment (metastatic castration-resistant prostate cancer, mCRPC). We found that certain lipids were consistently associated with poorer clinical outcome, while cytokines were not. The levels of a type of lipid (ceramide) were associated with some cytokines. This lipid is known to activate the immune system and is associated with poor outcomes in mCRPC. A change in lipid profiles was associated with better response to treatment. Overall, our findings suggest that blood lipids might be more informative than cytokines, might influence the immune response, and might help predict treatment response.

**Abstract:**

Circulating lipids or cytokines are associated with prognosis in metastatic castration-resistant prostate cancer (mCRPC). This study aimed to understand the interactions between lipid metabolism and immune response in mCRPC by investigating the relationship between the plasma lipidome and cytokines. Plasma samples from two independent cohorts of men with mCRPC (*n* = 146, 139) having life-prolonging treatments were subjected to lipidomic and cytokine profiling (290, 763 lipids; 40 cytokines). Higher baseline levels of sphingolipids, including ceramides, were consistently associated with shorter overall survival in both cohorts, whereas the associations of cytokines with overall survival were inconsistent. Increasing levels of IL6, IL8, CXCL16, MPIF1, and YKL40 correlated with increasing levels of ceramide in both cohorts. Men with a poor prognostic 3-lipid signature at baseline had a shorter time to radiographic progression (poorer treatment response) if their lipid profile at progression was similar to that at baseline, or their cytokine profile at progression differed to that at baseline. In conclusion, baseline levels of circulating lipids were more consistent as prognostic biomarkers than cytokines. The correlation between circulating ceramides and cytokines suggests the regulation of immune responses by ceramides. The association of treatment response with the change in lipid profiles warrants further research into metabolic interventions.

## 1. Introduction

Androgen-deprivation therapy (ADT) is the standard of care for the initial diagnosis of metastatic prostate cancer, as prostate cancer growth relies on testosterone and ADT lowers circulating testosterone levels to castrate levels [1]. When prostate cancer develops resistance to testosterone suppression, the disease is referred to as metastatic castration-resistant prostate cancer (mCRPC). Patients continue taking ADT while undergoing additional treatment for mCRPC. ADT can cause adverse cardiovascular or metabolic effects such as decreased insulin sensitivity, increased levels of blood cholesterol and triglycerides, decreased lean mass, and increased fat mass [2].

In the past decade, the therapeutic landscape of mCRPC has shifted from docetaxel chemotherapy as the only effective standard of care to multiple life-prolonging options that are clinically approved. These include targeted agents such as the novel androgen receptor signalling inhibitors (ARSI) (e.g., abiraterone and enzalutamide), lutetium-177-prostate-specific membrane antigen, and PARP inhibitors, as well as another taxane, cabazitaxel [1,3]. However, the long-term control of potentially lethal metastatic prostate cancer requires strategies targeting the many hallmarks of cancer that incorporate the neoplastic epithelium, the tumour microenvironment, immune response, and systemic metabolic factors (e.g., lipid metabolism), as all these promote cancer growth and treatment resistance [4].

Evidence for the role of dysregulated lipid metabolism in the clinical outcomes of metastatic prostate cancer is increasing [5]. Obesity is associated with higher rates of relapse and prostate-cancer-specific mortality [6]. Circulating lipid profiles that are rich in sphingolipids, especially ceramides, are associated with higher rates of metastatic relapse in prostate cancer, shorter time to androgen-deprivation therapy failure in metastatic hormone-sensitive prostate cancer (mHSPC), and shorter overall survival (OS) in mCRPC [7,8]. Furthermore, a poor prognostic three-lipid signature (3LS) in the plasma of patients with mCRPC was validated in two independent cohorts [7,8]. It is important to note that the association of circulating ceramides with prognosis in both castration (mCRPC) and non-castration settings (localised and mHSPC) indicates that the prognostic association of these lipids was independent of the metabolic effects of ADT.

Circulating lipids may be contributing to prostate cancer progression through the modulation of the immune response [9,10]. For example, ceramide can be metabolised into sphingosine-1-phosphate (S1P), which mediates innate and adaptive immunity by binding to specific G-protein-coupled receptors [11]. Mice deficient in the transporter for S1P develop less metastases when injected with tumour cells compared to wildtype mice [12]. A high-fat diet increased S1P production in tumours in mice models of breast cancer and promoted tumour progression, which could be blocked by pharmacological inhibition of S1P signalling [13]. Ceramide can also activate the myeloid cell receptor, CD300f, which negatively regulates dendritic-cell-initiated T-cell responses [14]. The inhibition of CD300f enhanced the anti-tumour effect of immunisation in a mouse model [15].

Several lines of evidence indicate an interplay between the immune response and lipid metabolism in prostate cancer progression. For example, a high fat diet was sufficient to drive metastasis in the non-metastatic Pten-null mouse model of prostate cancer [16]. Additionally, the inhibition of de novo lipogenesis suppressed androgen receptor signalling and CRPC growth in xenograft and organoid models [17]. However, the growth of early-stage prostate cancer patient-derived xenografts in immunodeficient mice were unaffected by a high fat diet [18]. Furthermore, high-fat-diet-fed Pten-null mice exhibit increased tumour growth through increased myeloid-derived suppressor cells and IL6-STAT3 signalling; and tumour growth was inhibited by administration of IL6 receptor antibody [19].

The role of IL6 and other cytokines in driving prostate cancer progression have been demonstrated by studies with cell lines and mouse models [20,21,22,23]. The prognostic value of circulating cytokines has also been demonstrated for prostate cancer. For example, higher levels of serum IL8 in men with metastatic hormone-sensitive prostate cancer (mHSPC) at initiation of androgen-deprivation therapy was associated with shorter survival and time to castration-resistant disease [24]. An increase in circulating IL6 after a cycle of docetaxel in men with mCRPC was associated with progressive disease [25]. However, few biomarker studies investigated the prognostic association of circulating lipids together with the immune response. These studies mainly examined the blood levels of triglycerides and cholesterol with a few inflammatory markers (e.g., C-reactive protein) or tissue inflammation [26,27,28].

Studies on the interaction between the immune response and lipid metabolism rely on in vitro and animal models, whereas the tumour microenvironment and host metabolism in humans are more heterogenous and complex. It is essential to understand how these different biological systems interact within human cohorts. Therefore, our study aims to examine the relationship between the plasma lipidome and multiple cytokines in men with mCRPC.

## 2. Materials and Methods

### 2.1. Patients and Plasma Collection

Plasma samples used in this study were from two cohorts of patients with mCRPC in Australia, referred to as Cohort 1 and Cohort 2. Cohort 1 consisted of 146 patients from the study described by Lin et al. (2017) [8] and was recruited from August 2006 to July 2015 from seven hospitals in New South Wales—Chris O’Brien Lifehouse, Concord Repatriation General Hospital, Mid North Coast Cancer Institute, Sydney Adventist Hospital, Westmead Hospital, Calvary Mater Hospital, St. George Hospital. Cohort 2 was independently recruited from June 2016 to February 2020 from the first four hospitals as Cohort 1, St. Vincent’s Hospital in Sydney, and three hospitals in Victoria (Epworth HealthCare, Monash Health, and Eastern Health).

Blood was collected in EDTA-containing tubes from the participants prior to commencing a new line of treatment, according to a standardised blood collection protocol that differed in the plasma separation technique for both cohorts. For Cohort 1, the blood was centrifuged at 3000 g for 5 min, and the separated plasma transferred to a fresh tube and stored as aliquots at −70 °C. For Cohort 2, the blood was centrifuged at 1600× *g* for 15 min, and the separated plasma was transferred to a fresh tube and centrifuged again at 5000× *g* for 10 min. Aliquots of the plasma after the second spin were then stored at −70 °C.

Clinical data for Cohort 1 from the first study [8] were updated with new follow-up data, as described by Lin et al. (2021) [7]. Cohort 2 was seen by their clinician every two months. Overall survival (OS), radiographic progression (detection of metastases by radiography or imaging, e.g., bone scan), or serum prostate-specific antigen (PSA) progression (rise in PSA levels) were used as measures of clinical outcomes. Radiographic and PSA progression were defined as by the Prostate Cancer Clinial trials Working Group (PCWG2) [29]. Radiographic progression-free survival (rPFS), PSA progression-free survival (PSA-PFS), and OS were defined as the time from commencement of treatment line to the date of the event, and censored at the date of last follow-up if the event has not occurred.

### 2.2. Lipidomic Analysis

Lipidomic analysis of plasma samples from Cohort 1 was described in Lin et al. (2017). In that study, the plasma samples were analysed as two separate cohorts and included samples from 13 other patients. The normalised lipidomic datasets from that study were aligned using the ComBat algorithm [30], resulting in 290 lipids in common as reported in Lin et al. (2021) [7]. Lipidomic analysis of plasma samples from Cohort 2 was performed using the methodology described in Huynh et al. (2019) [31]. A total of 763 lipid species were measured. The lipidomic data was normalised using the Probabilistic Quotient Normalisation method, with the reference created by the average of each lipid species [32]. The final normalised levels of lipids were transformed to log2 scale for statistical analyses.

### 2.3. Cytokine Profiling

Multiplex analysis of cytokines in plasma samples from Cohort 1 and Cohort 2 were analysed separately, 2 years apart, using the following Milliplex Map assays (EMD Millipore Corporation, Billerica, MA, USA) according to the manufacturer’s instructions with the Bio-Plex MAGPIX system (Biorad Laboratories, Australia, #171015044) and Bio-Plex Pro-Wash Station (Biorad)—Human High Sensitivity T Cell Magnetic Bead Panel (21-plex pre-mixed, 18 May 2017 edition), Human Th17 Magnetic Bead Panel (10-plex, 18 May 2017 edition), and Human Cytokine/Chemokine Panel IV Magnetic Bead Panel (18 May 2017 edition [21-plex pre-mixed] for Cohort 1, 18 November 2019 edition [20-plex pre-mixed] for Cohort 2). Cohort 1 samples were analysed in two batches, whereas Cohort 2 samples were analysed in three batches. The same sample layout on the multi-well plate was used for each type of assay. Data was generated using the Bio-Plex Manager MP and Bio-Plex Manager 6.1 software. Cytokine levels were exported as the observed concentration and converted to ng/mL.

The datasets from both cohorts were pre-processed separately as follows. Cytokines with more than 50% missing values (zero or below range) were removed. Missing values of remaining cytokines were replaced with half of the lowest value for that cytokine. Concentrations for values above range (three samples in Cohort 2) were estimated from interpolation of their standard curve using Prism (version 8.4.3, GraphPad software; method: Sigmoidal, 4PL, X is log concentration, least squares fit). Next, each batch of samples was normalised by the Probabilistic Quotient Normalisation method, using the average of each cytokine to create the reference, and the final values transformed into log2 scale. Finally, the different batches were combined with batch correction performed with the ComBat algorithm [30] (R package: sva, v3.33.1). Statistical analyses were performed with values in log2 scale. A total of 40 cytokines were in common between Cohort 1 and 2 after the data pre-processing step, and this number of cytokines was used in the statistical analyses.

### 2.4. Statistical Analysis

Statistical analyses were performed with R (version 3.6.0) using the R packages listed accordingly. *p*-values < 0.05 were considered as statistically significant. The association of lipid or cytokine levels with OS, rPFS, or PSA-PFS was determined by Cox regression as continuous variables unless stated otherwise (survival, v2.44-1.1).

Plasma samples of Cohort 2 were classified by the poor prognosis three lipid signature (3LS) that was originally derived in the study involving Cohort 1 by Lin et al. (2017) [8]. First, the lipidomic dataset of Cohort 2 was aligned to the original lipidomic dataset of that study (Phase 1) using the ComBat algorithm [30] (sva, v3.33.1) to remove batch differences, as the lipidomic datasets were produced by different LC-MS instruments and on a different occasion. Next, the 3LS status of each plasma sample was calculated from a logistic regression model consisting of ceramide Cer(d18:1/24:1), sphingomyelin SM(d18:2/16:0) and phosphatidycholine PC(16:0/16:0), derived in Lin et al. (2017) as follows [8]:y = (3.1319 × Cer(d18:1/24:1)) + (2.1724 × SM(d18:2/16:0)) + (1.8593 × PC(16:0/16:0)) − 91.217
*p* = e^y^/(1 + e^y^)
Patient has the poor prognostic 3LS when *p* ≥ 0.5

Enrichment of lipid types was determined by one-sided Fisher’s exact tests using bc3net (v1.04). Kaplan–Meier curves were drawn with survminer (v0.4.6). Pearson correlation analysis between lipid and cytokine levels was performed with Hmisc (v4.2-0). Heatmaps and hierarchical clustering (complete linkage) were produced with pheatmap (v1.0.12).

## 3. Results

### 3.1. Study Cohorts and Plasma Samples

The cohorts and study schema are summarised in Figure 1. Cohort 1 has been described previously with regards to their lipidomic analysis [7] and consists of 146 men who received docetaxel as a first line therapy for mCRPC. Cohort 2 is an independent cohort of 139 men who received first line or subsequent life-prolonging therapy for mCRPC.

For all comparisons to Cohort 1, a subset of Cohort 2 (Cohort 2a, *n* = 128 men) was studied as baseline plasma samples from first line therapy were available from these men. The clinical characteristics of Cohorts 1 and 2a are summarised in Table 1. The majority (74%) of Cohort 2a received enzalutamide or abiraterone as first line. There was a high rate of subsequent life-prolonging therapies after the first line treatments in both cohorts (Table 1).

In order to analyse the changes in lipid and cytokine profiles at progression on therapy, another subset of Cohort 2, referred to as Cohort 2b, was examined. Cohort 2b consisted of 51 men with matched baseline and end of treatment (EOT) plasma samples. The EOT samples were collected at disease progression or at baseline of the subsequent treatment (Figure 1). There were 56 pairs of baseline EOT samples, as 5 men had paired samples from 2 subsequent treatment lines, whereas the rest only had matched samples from 1 treatment line.

### 3.2. Circulating Lipids Associated with Clinical Outcomes

In our previous study involving Cohort 1 [8], we had derived a poor prognostic three-lipid signature (3LS) to represent the poor prognostic lipidomic profiles which have higher plasma levels of ceramides. The 3LS is comprised of ceramide Cer(d18:1/24:1), sphingomyelin SM(d18:2/16:0), and phosphatidylcholine PC(16:0/16:0). Using additional follow-up data, we confirmed that the men in Cohort 1 who have the 3LS at baseline of the first line therapy have shorter OS (Hazard ratio (HR) = 3.70, 95% CI = 2.40–5.70, logrank *p* < 0.0001, Figure 2a) and PSA-PFS (HR = 1.90, 95% CI = 1.26–2.86, logrank *p* = 0.002, Appendix A). The rPFS information was not available for Cohort 1, hence, PSA-PFS was analysed instead as an indicator of disease progression. The presence of the 3LS at baseline of the first line therapy of Cohort 2a was confirmed to also be associated with shorter OS (HR = 1.81, 95% CI = 1.15–2.85, logrank *p* = 0.009, Figure 2a) and rPFS (HR = 2.01, 95% CI = 1.24–3.26, logrank *p* = 0.004, Appendix A).

Cox regression analyses of baseline levels of individual lipids for first line therapy identified 80 lipids for Cohort 1 and 128 lipids for Cohort 2a that were significantly associated with OS (*p* < 0.05, Figure 2b, Appendix A). There were 26 prognostic lipids that were in common between both cohort (Appendix A). Lipids with higher baseline levels that were associated with shorter OS were significantly enriched with sphingolipids such as ceramide (Cer(d)), trihexosylceramide (Hex3Cer), and ganglioside GM3 for both cohorts and sphingomyelin for Cohort 1 (Fisher’s exact test *p* < 0.05, Appendix A). Lipids with lower levels that were associated with shorter OS in Cohort 1 were enriched for TG and LPC, whereas those in Cohort 2a were enriched for the ether lipids PE(O), PE(P) and TG(O), and PC (Fisher’s exact test *p* ≤ 0.04, Appendix A).

Higher levels of free cholesterol were associated with shorter OS in both cohorts (Cohort 1, HR = 4.95, 95% CI = 1.85–13.3, *p* = 0.0015; Cohort 2a, HR = 3.42, 95% CI = 1.25–9.32, *p* = 0.016; Figure 2b). However, the association of cholesteryl esters (CE) was mixed in both cohorts (Figure 2b). The levels of total cholesterol, which is the sum of free cholesterol and CE, was not associated with OS in either cohort (Cohort 1, HR = 1.27, 95% CI = 0.51–3.16, *p* = 0.60; Cohort 2a, HR = 1.35, 95% CI = 0.60–3.04, *p* = 0.46).

The baseline levels of 115 lipids of Cohort 2a was associated with rPFS (Cox regression *p* < 0.05, Appendix A), of which 62 lipids were in common with those that were associated with OS (Figure 2c). Of these 62 lipids, most of the lipids with elevated levels associated with shorter rPFS and OS were sphingolipids (13 (65%) of 20), whereas most of those with decreased levels associated with shorter rPFS and OS were ether lipids (37 (88%) of 42) (Figure 2c).

### 3.3. Circulating Cytokines Associated with Clinical Outcomes

In contrast, the baseline levels of only a few cytokines were associated with OS or rPFS. In Cohort 1, higher levels of IL8 were associated with shorter OS (HR = 1.22, 95% CI = 1.04–1.45, *p* = 0.02, Figure 3a, Table 2, Appendix A), which was not validated in Cohort 2a (*p*= 0.29, Figure 3a). Instead, higher levels of IL6, MIP1b, ITAC, and YKL40 were associated with shorter OS for Cohort 2a (Cox regression *p* ≤ 0.048, Figure 3a, Table 2, Appendix A). There were no cytokines with baseline levels associated with rPFS in Cohort 2a (Cox regression *p* ≥ 0.05, Appendix A), although IL6 almost reached statistical significance with *p* = 0.053 (HR = 1.16, 95% CI = 0.998–1.34).

### 3.4. Combined Association of Prognostic Lipid Signature and Cytokines with OS

The next step was to assess the combination of lipids and cytokines in relation to OS using the poor prognostic 3LS to represent the poor prognostic lipid profile. A bivariable Cox regression of the 3LS with any of the 40 cytokines showed that the 3LS was always independently associated with OS in both cohorts (Cox regression *p*-value ranged from <0.001 to 0.02, Appendix A). Only a few cytokines were independently associated with OS in bivariable analysis with the 3LS – IL27 or IL5 for Cohort 1 and IL6 or MIP1b for Cohort 2a (*p* < 0.05, Table 2). Of all the cytokines that were significantly associated with OS in univariable analysis for either cohorts (Figure 3a), only IL6 and MIP1b were associated with OS in bivariable analysis with the 3LS (Table 2).

In Cohort 1, men with the 3LS at baseline had shorter OS if they also had high levels (>median) of either IL6, CXCL16, IL27, or CCL28 compared to men who had the 3LS together with lower levels (<median) of any of these cytokines (logrank *p* < 0.05, Figure 3b). However, these findings were not recapitulated in Cohort 2a. Instead, men with the 3LS at baseline in Cohort 2a had shorter OS if they had high levels (>median) of either IL15, YKL40, or MIP1b (logrank *p* < 0.05) compared to men who had the 3LS together with low levels (<median) of any of these cytokines (Figure 3c).

In summary, the 3LS is consistently prognostic and independent of the cytokines in both cohorts, whereas the prognostic association of cytokines was inconsistent.

### 3.5. Correlation between Lipids and Cytokines

Despite the lack of prognostic efficacy for the cytokines, a key question remains around the biological interaction between the lipid and the cytokine profiles. To this end, we analysed the correlation between the levels of the lipids and cytokines. For this analysis, we focused on the 26 lipids that were associated with OS in both Cohort 1 and Cohort 2a and examined the correlation of the levels of these lipids with cytokines in all the plasma samples of Cohort 1 or Cohort 2 (Figure 4, Appendix A). Increasing levels of the poor prognostic ceramides in both cohorts were weakly correlated with increasing levels of MPIF1, IL6, IL8, CXCL16, and YKL40 (Pearson coefficient = 0.14 to 0.43, *p* <0.0001 to 0.049, Figure 4, Appendix A). The increasing levels of these cytokines were also weakly correlated with increasing levels of some of the other poor prognostic sphingolipids such as the hexosylceramides (HexCer, Hex2Cer, Hex3Cer) or gangliosides (Pearson coefficient = 0.15 to 0.35, *p* < 0.0001 to 0.045, Figure 4, Appendix A).

In Cohort 1, decreasing levels of interferon-γ (IFNg), IL17a, IL1B, and IL2 correlated with increasing levels of ceramides (Pearson coefficient = −0.17 to −0.28, *p* = 0.0005 to 0.041, Figure 4, Appendix A). However, in Cohort 2, interferon-γ and IL1B did not correlate with any of the poor prognostic ceramides; decreasing levels of IL17a and IL2 correlated with increasing levels of only half of the poor prognostic ceramides (Figure 4, Appendix A).

Neutrophil to lymphocyte ratio (NLR) is a known marker of poor prognosis in cancer, and comprehensive data of this was only available for Cohort 2. A high NLR was associated with shorter OS in Cohort 2a (HR = 1.13, 95% CI = 1.06–1.20, *p* = 0.0002). The correlation analyses of 121 prognostic lipids or cytokines with NLR showed that only 21 lipids and 2 cytokines were correlated with NLR, of which only one of them is a sphingolipid—a ganglioside (Pearson coefficient = 0.19, *p* = 0.03, Appendix A).

### 3.6. Changes in Lipid and Cytokine Profiles during Treatment

To investigate the changes in lipid and cytokine profiles in individual patients over time as they become resistant to treatment and start new lines of therapy, paired baseline and EOT plasma samples from Cohort 2b were analysed (Figure 1). There were 56 pairs of matched baseline and EOT samples, with 36 from ARSI therapy, 19 from docetaxel therapy, and 1 from cabazitaxel therapy.

Principal components analysis (PCA) of the EOT samples by the profiles of 121 prognostic lipids or 40 cytokines showed that the EOT lipid or cytokine profiles do not group together by treatment (Appendix A). The hierarchical clustering of the 56 pairs of baseline and EOT plasma samples according to the profiles of 121 prognostic lipids or 40 cytokines revealed that the matched samples from the same individual tend to pair up together instead of clustering with other samples by treatment (Figure 5, Appendix A). Of the 56 pairs of baseline EOT samples, 32 pairs (57%) paired together according to their lipid profile, while 35 pairs (62%) paired together according to their cytokine profile. The clustering of a baseline sample next to its matched EOT sample is an indication that their profiles are very similar.

To determine if the similarity of lipid or cytokines profiles at baseline and EOT were related to treatment response, the rPFS of treatment lines with the poor prognostic 3LS at baseline were examined. There were 16 pairs of matched baseline EOT samples with the 3LS at baseline, of which matched samples with similar lipid profiles tend to have different cytokine profiles, and vice versa (Figure 6a). Treatment lines with the 3LS at baseline have shorter rPFS if the lipid profiles at baseline and EOT were similar (median rPFS = 3.8 versus 8.3 months, logrank *p* = 0.02, Figure 6b). In contrast, treatment lines with the 3LS at baseline have shorter rPFS if the cytokine profiles at baseline and EOT were different (median rPFS = 3.6 vs. 8.3 months, logrank *p* = 0.03, Figure 6c). Thus, lipid profiles with the 3LS at baseline that remained similar at EOT have poorer treatment response, whereas cytokine profiles that changed at EOT have poorer treatment response.

The examination of the difference in the levels of the 121 prognostic lipid levels at EOT from baseline did not reveal any remarkable differences among lipids of the same type, although the levels of TG(O) species tend to be lower at EOT for lipid profiles that differed at EOT from baseline (*t*-test *p* ≥ 0.05, Figure 6d). However, for those cytokine profiles that differed at EOT from baseline, the levels of IL8 were significantly higher at EOT (1.4-fold, paired *t*-test *p* = 0.002), and the levels of three T-helper-17 (Th17) cytokines (IL17E, IL28A, IL33) were lower at EOT (0.38–0.63-fold, paired *t*-test *p* ≤ 0.04) (Figure 6e). The levels of five other Th17 cytokines were also lower at EOT but the difference was not statistically significant (*p* ≥ 0.05, Figure 6e).

## 4. Discussion

This study identified a significant relationship between circulating lipids and cytokines in mCRPC. We confirmed that increased levels of circulating sphingolipids, particularly ceramides, were associated with poor clinical outcomes in another independent cohort of mCRPC (Cohort 2a). The association between the baseline levels of cytokines and OS were inconsistent in both cohorts, indicating that circulating lipids perform better as prognostic biomarkers than the cytokines. However, although the cytokines do not perform as well as lipids, the cytokines are still important in prostate cancer progression as other studies provide evidence on their pathological role in prostate cancer, and the levels of some cytokines were correlated with that of the ceramides (e.g., IL6 and IL8) in the two independent cohorts.

The correlation between certain cytokines and prognostic ceramides suggests that there may be an interplay between the cytokines and lipids. Possible scenarios would be induction of cytokine expression by ceramides directly or indirectly in immune cells, cancer cells or other cell types, which could be occurring in the tumour stroma, circulation, or organs. Alternatively, perhaps the relationship between cytokines and ceramides are independent, where the weak correlation exists simply from a weak prognostic relationship between cytokines and clinical outcome. In the following paragraphs, we present the existing knowledge on the connection between ceramides and cytokines.

In both cohorts of our study, elevated levels of IL6, IL8, CXCL16, and YKL40 were consistently significantly correlated with elevated levels of ceramides. Other studies have shown that elevated circulating levels or tumour expression of these four cytokines were associated with poorer clinical outcomes in prostate cancer [21,22,23,24,33,34]. In vitro or in vivo models demonstrated that IL6, IL8, and CXCL16 can promote prostate cancer growth or metastasis [20]. In our study, elevated plasma levels of IL6, IL8, and YKL40 were associated with shorter OS in only one cohort. To date, only the production of IL6 and IL8 have been connected to ceramides.

Increasing plasma levels of IL6 was correlated to increasing plasma levels of ceramide in patients with cardiovascular disease [35]. IL6 expression in human fibroblasts was induced by treatment with ceramide or sphingomyelinase which hydrolyses sphingomyelin into ceramide [36]. The inhibition or knockdown of sphingomyelinase in cancer or immune cells prevented IL6 production [37,38,39], whereas the promotion of cellular ceramide levels by increasing sphingomyelinase expression in bladder cancer cells enhanced IL6 production [39]. IL6 expression by breast cancer metastases was blocked by inhibition of sphingosine kinase, which produces sphingosine-1-phosphate (S1P) from ceramide [13]. Myeloid cells or murine bladder cancer cells with elevated expression of S1PR1 (one of the receptors for S1P) produced more IL6 [40]. Circulating levels of IL6 in mice was reduced by treatment with FTY720, an antagonist against S1PR1 [41]. The induction of IL8 by S1P was reported in epithelial, endothelial, and smooth muscle cells [42,43,44,45]. Circulating levels of S1P was not associated with OS in our study. However, circulating S1P levels were weakly correlated to that of ceramides (Pearson correlation = 0.26 to 0.30, *p* ≤ 0.003, Appendix A). Furthermore, S1P has a short half-life and extremely low cellular levels (ceramide:S1P ratio = 3000:1) [46,47].

Other cytokines that may be regulated by ceramide are IFNγ and IL2, which are typically associated with Th1 cell responses [48]. Th1 cells are more often linked to an anti-tumour immune response, as their intratumoral presence or derived cytokines were associated with improved outcomes across multiple tumour types [48,49]. In Cohort 1, we observed that decreasing levels of IFNγ and IL2 correlated with increasing levels of ceramides. Ceramides may be obstructing Th1 cell responses by activating the myeloid cell receptor CD300f, as the inhibition of CD300f promoted dendritic-cell-initiated Th1 responses when enhancing the anti-tumour effect of immunisation in a mouse model [15].

The independent relationship of circulating lipids and cytokines in our study was demonstrated by the association of treatment response with the overall profiles of prognostic lipids and cytokines at radiographic progression, for patients with the poor prognostic 3LS at the start of treatment. The prognostic lipid profiles with the 3LS at baseline have shorter rPFS if their lipid profile at progression (EOT) was similar to that at baseline or if their cytokine profile at progression was different to that at baseline. Specific lipid types with significant changes that relate to a better treatment response could not be identified and are likely to involve the combined minute effects of multiple lipids. A poorer treatment response was associated with an increase in the levels of IL8 and the decrease in the levels of several Th17 cytokines at EOT. In this situation, the relationship between IL8 and Th17 response appears to be independent, given that IL17 can stimulate cells to produce IL8 [50,51,52]. On the other hand, the decrease in Th17 cytokines with treatment response is consistent with Th17′s ability to mediate anti-tumour immunity through the promotion of Th1 responses [52,53,54]. The role of Th17 response in tumor immunity is conflicting and likely to be context dependent [52]. The findings from our study imply that metabolic intervention that induce a change in the poor prognostic lipid profile may improve outcome.

We also found that higher levels of free cholesterol were associated with shorter OS in both cohorts, although total cholesterol was not. Other studies have reported that total cholesterol levels in patients undergoing radical prostatectomy were not associated with recurrence, except for men with dyslipidemia [55,56]. However, statin usage was associated with better clinical outcomes in localised and metastatic prostate cancer [57,58,59]. It is unclear if the beneficial effect of statin is due to its cholesterol-lowering properties alone, as statins also decreased circulating levels of ceramides [60,61].

## 5. Conclusions

In summary, higher levels of circulating sphingolipids including ceramides were associated with shorter OS in mCRPC and were correlated with elevated pro-inflammatory cytokine levels across two independent patient cohorts. Circulating lipids appeared to be better prognostic biomarkers than cytokines, as the baseline levels of the lipids were more consistently associated with OS. These data support the concept of the circulating lipid profile as a potentially druggable target, which may also affect the immune environment downstream.

## Figures and Tables

**Figure 1 cancers-13-04964-f001:**
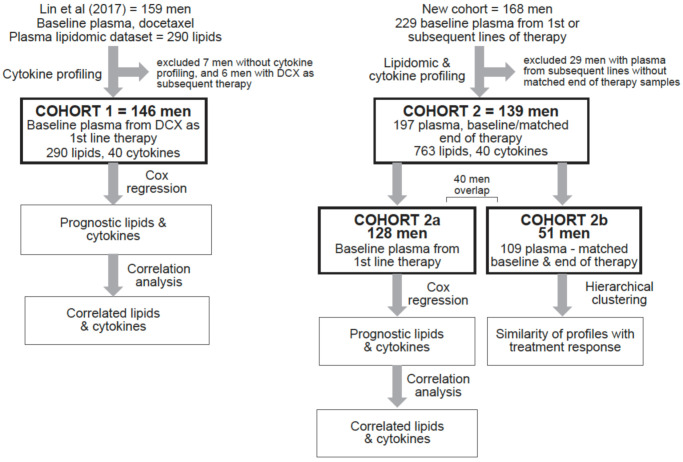
Study cohorts and schema (DCX, docetaxel).

**Figure 2 cancers-13-04964-f002:**
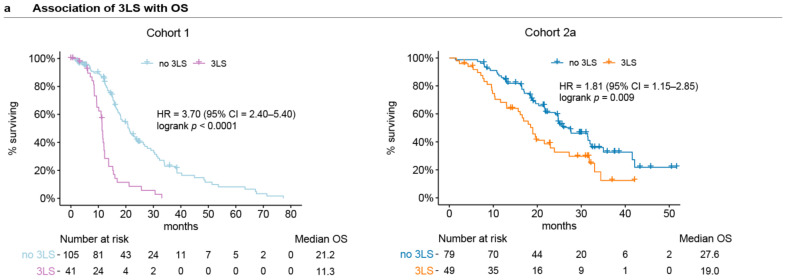
Prognostic baseline levels of circulating lipids for Cohort 1 and Cohort 2a: (**a**) Kaplan–Meier curves of the association of the poor prognostic 3-lipid signature (3LS) at baseline with OS. The 3LS consists of Cer(d18:1/24:1), SM(d18:2/16:0), and PC(16:0/16:0); (**b**) Forest plots of hazard ratios of OS for baseline plasma levels of lipids in both cohorts; (**c**) Forest plot of hazard ratios of rPFS for baseline plasma levels of lipids significantly associated with rPFS and OS in Cohort 2a.

**Figure 3 cancers-13-04964-f003:**
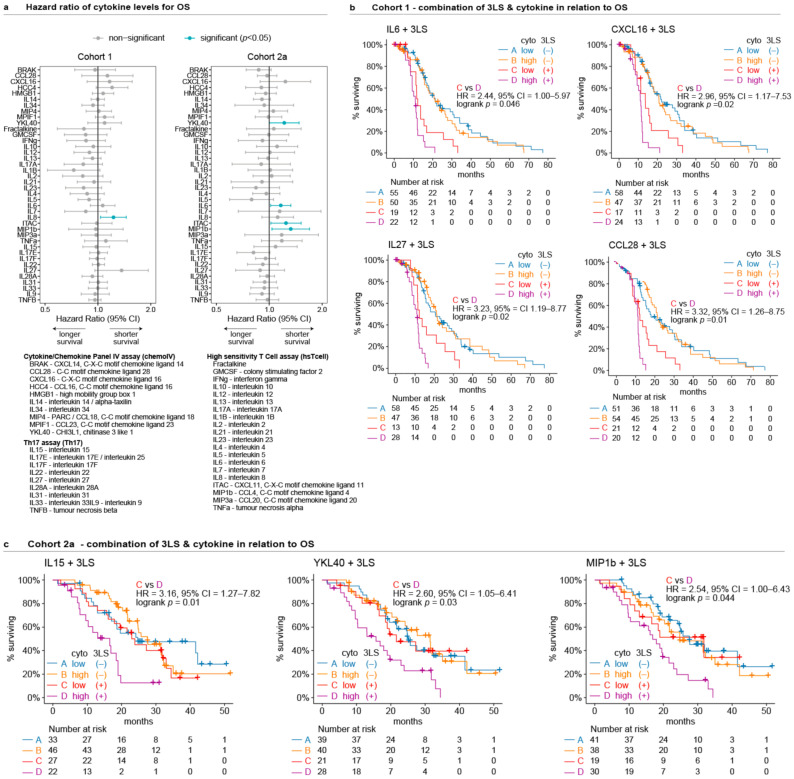
Prognostic baseline levels of circulating cytokines for Cohort 1 and Cohort 2a: (**a**) Forestplots of hazard ratios for baseline plasma levels of cytokines in relation to OS; (**b**,**c**) Kaplan–Meier curves of the combined presence of 3LS with cytokine levels in relation to OS in Cohorts 1 and 2a, respectively (high cytokine level = above median, low cytokine level = below median).

**Figure 4 cancers-13-04964-f004:**
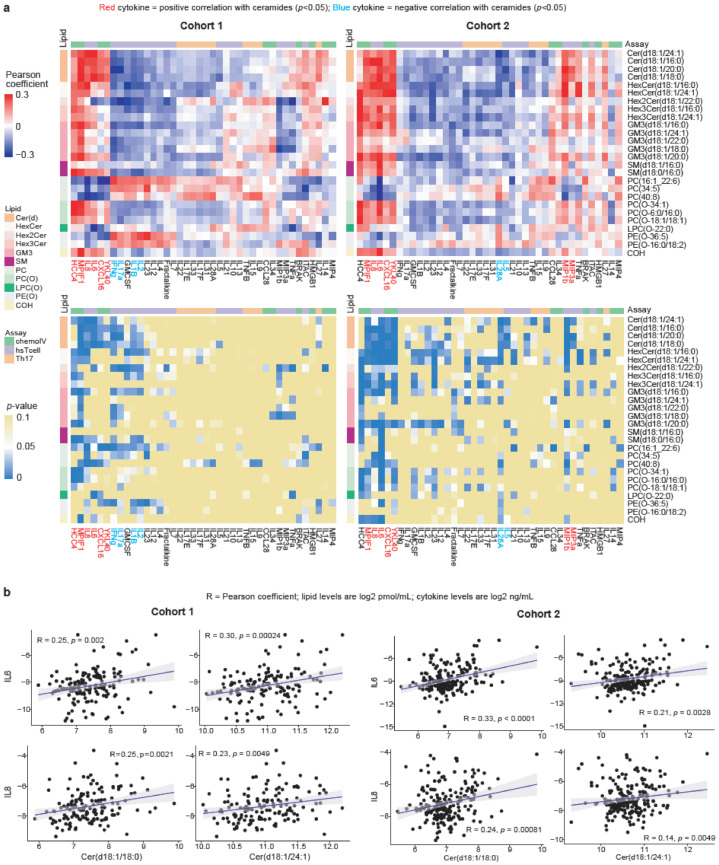
Correlation between circulating lipids and cytokines: (**a**) Heatmaps of Pearson coefficient and *p*-values of correlation between the plasma levels of cytokines and 26 prognostic lipids for Cohort 1 (left, 146 plasma samples) and Cohort 2 (right, 197 plasma samples). The cytokines are ordered by the hierarchical clustering of the Pearson coefficients of Cohort 1 (Euclidean distance); (**b**) Examples of scatterplots of the plasma levels IL6 or IL8 versus those of ceramide(d18:1/18:0) or ceramide(d18:1/24:1).

**Figure 5 cancers-13-04964-f005:**
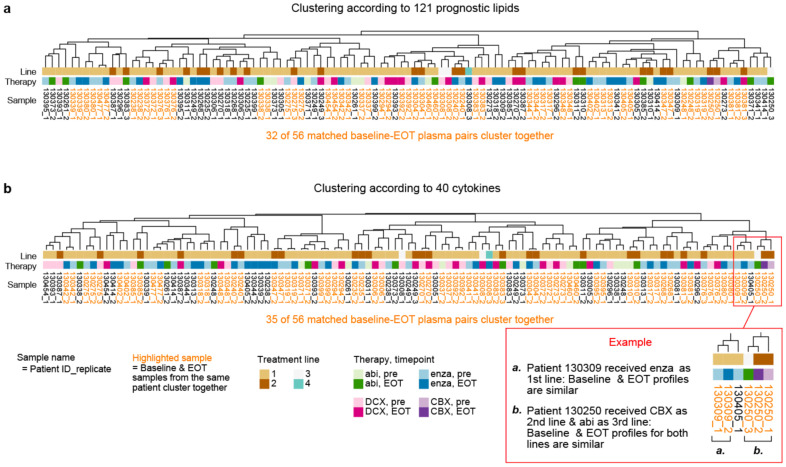
Hierachical clustering of 56 pairs of matched baseline and EOT plasma samples for Cohort 2b. Samples are clustered according to the profiles of 121 prognostic lipids (**a**) or 40 cytokines (**b**). Matched baseline and EOT samples that paired together (cluster next to each other) are highlighted in orange. Heatmaps of the lipid or cytokine levels are displayed in Appendix A.

**Figure 6 cancers-13-04964-f006:**
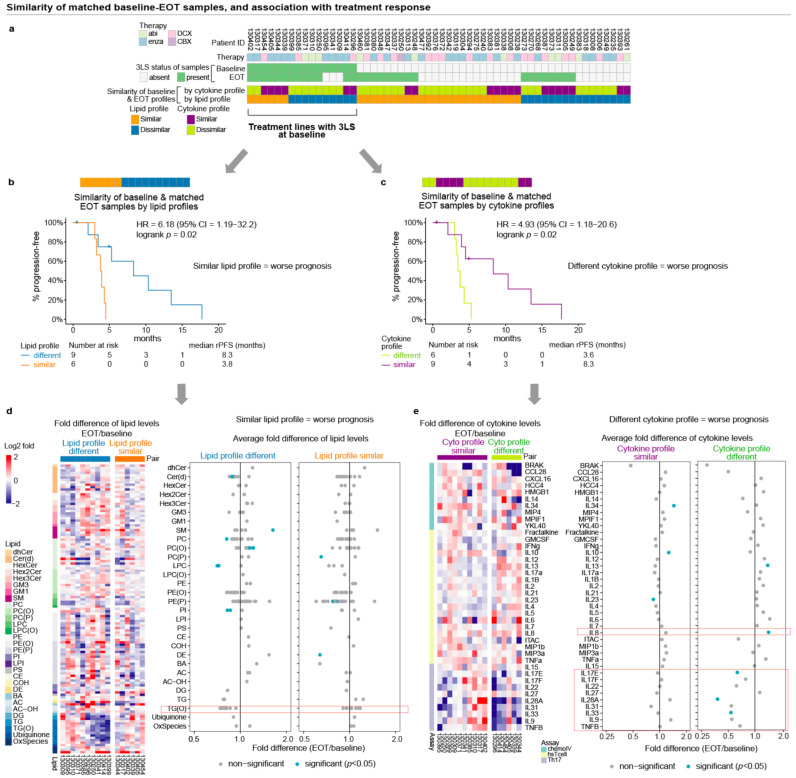
Association of prognostic lipid profiles with treatment response in Cohort 2b: (**a**) Status of 3LS and similarity of matched baseline and EOT samples according to hierarchical clustering from Figure 5; (**b**,**c**) Kaplan–Meier curves of rPFS for treatment lines with 3LS at baseline, stratified by similarity of baseline EOT lipid or cytokine profiles; (**d**,**e**) lipid and cytokine fold change at EOT displayed as heatmap of fold change for each baseline EOT pair and forest plot of average fold change. Lipids/cytokines with notable changes are indicated by the orange outline in the forest plot.

**Table 1 cancers-13-04964-t001:** Clinical characteristics of Cohorts 1 and Cohort 2a (Q1, first quartile; Q2, second quartile).

Characteristic	Cohort 1	Cohort 2a
Number of men	146	128
Plasma collection period	August 2006–July 2015	June 2016–February 2018
Follow-up time, median (Q1, Q3) (months)	14.2 (8.7, 23.4)	19.6 (12.9, 28.0)
Deaths	113 (77%)	77 (60%)
Radiographic progression	n.a	67 (52%) (1 n.a.)
PSA progression	116 (79%) (8 n.a.)	90 (70%)
Age, baseline, median (Q1, Q3) (years)	70.2 (64.3, 75.0)	75.2 (67.8, 81.4)
PSA, baseline, median (Q1, Q3) (ng/mL)	113 (39.3, 394)	27.9 (10.7, 60.6)
Alkaline phosphatase, baseline, median (Q1, Q3) (U/L)	134 (92, 301)	113 (78.3, 175.3)
Hemoglobin, baseline, median (Q1, Q3) (g/L)	124 (109.5, 134.5)	129 (119,139)
Gleason, baseline		
≤6	10 (6.8%)	6 (4.7%)
7	32 (22%)	24 (19%)
8–10	72 (49%)	77 (60%)
No info	32 (22%)	21 (16%)
Metastasis, baseline		
None	3 (2%)	15 (12%)
Bone only	83 (57%)	104 (81%)
Visceral only	8 (5%) *	3 (2%)
Both	27 (18%) *	6 (5%)
Unknown	8 (5%)	
1st line treatment		
Enzalutamide	0	65 (51%)
Abiraterone	0	29 (23%)
Docetaxel	146 (100%)	33 (26%)
Cabazitaxel	0	1 (1%)
Subsequent treatments		
Abiraterone	78 (53%)	16 (13%)
Enzalutamide	18 (12%)	15 (12%)
Docetaxel	5 (3%)	16 (13%)
Cabazitaxel	45 (31%)	19 (15%)
Mitoxantrone	15 (10%)	1 (1%)
Carboplatin	3 (2%)	4 (3%)
Lutetium-PSMA ± other	0	8 (6%)
Other	3 (2%)	10 (8%)

*, includes soft tissue; n.a., data not available

**Table 2 cancers-13-04964-t002:** Bivariable cox regression of the poor prognostic 3-lipid signature (3LS) with cytokines.

Cohort	Analysis	Variable *	Univariable Cox Regression		Bivariable Cox Regression	
			HR (95% CI)	*p*-Value	HR (95% CI)	*p*-Value
Cohort 1	Analysis 1	3LS	3.70 (2.40–5.70)	<0.001	3.50 (2.26–5.41)	<0.001
		IL8	1.23 (1.04–1.45)	0.018	1.17 (0.97–1.40)	0.096
	Analysis 2	3LS	3.70 (2.40–5.70)	<0.001	3.90 (2.51–6.05)	<0.001
		IL27	1.36 (0.96–1.94)	0.083	1.48 (1.04–2.13)	0.031
	Analysis 3	3LS	3.70 (2.40–5.70)	<0.001	3.93 (2.52–6.11)	<0.001
		IL5	0.89 (0.77–1.03)	0.12	0.86 (0.75–0.99)	0.041
Cohort 2	Analysis 1	3LS	1.81 (1.15–2.85)	0.009	1.73 (1.10–2.72)	0.019
		IL6	1.16 (1.02–1.33)	0.024	1.15 (1.00–1.31)	0.046
	Analysis 2	3LS	1.81 (1.15–2.85)	0.009	1.78 (1.13–2.80)	0.013
		MIP1b	1.33 (1.04–1.70)	0.025	1.30 (1.02–1.65)	0.033
	Analysis 3	3LS	1.81 (1.15–2.85)	0.009	1.73 (1.10–2.73)	0.018
		ITAC	1.25 (1.02–1.52)	0.030	1.22 (0.99–1.50)	0.058
	Analysis 4	3LS	1.81 (1.15–2.85)	0.009	1.76 (1.11–2.78)	0.016
		YKL40	1.22 (1.00–1.48)	0.048	1.18 (0.98–1.42)	0.077

* 3LS = absent vs. present; cytokine = continuous variable.

## Data Availability

Hazard ratios from Cox regression analyses of lipids and cytokines are provided in Appendix A. Pearson coefficients and *p*-values from correlation analyses are provided in Appendix A. Un-normalised lipid and cytokine concentrations are available upon request. Clinical information of individual patients cannot be provided due to the ethics restrictions.

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
