# Peer review of "Relationship between Circulating Lipids and Cytokines in Metastatic Castration-Resistant Prostate Cancer"

_cancers, 2021, doi:10.3390/cancers13194964_

Round 1

Reviewer 1 Report

The manuscript by Lin et al. entitled ” Relationship between circulating lipids and cytokines in meta-2 static castration-resistant prostate cancer” examined plasma lipidome and cytokines in two cohorts of metastatic castration-resistant prostate cancer patients. Although the data appear generally sound, some of the interpretations are not justified in the current version, and consequently, the conclusions are not justified.

The following points should be addressed  throughout (abstract, introduction, results and discussion).

  1. The proposed link between ceramide and immune response via S1P is tenuous, particularly as S1P is not significant in cohort 2a (Figure 2), which is different to ceramides.
  2. The assertion in the abstract (and results and discussion) that “correlation between circulating ceramides and cytokines supports the regulation of immune responses by ceramide-derived S1P” also doesn’t make sense as ceramide/lipids but not cytokines is associated with survival.
  3. In contrast, high cholesterol strongly correlated with shorter survival in both cohorts (Figure 2), and there are many publication on the role of cholesterol in prostate cancer prognosis. However, this lipid was not discussed at all.
  4. The 3-lipid signature are individual species, whereas most of the paper discuss lipid classes. Furthermore, one of the 3-LS, SM(d18:2/16:1) HR was not significant in cohort 2. Did the authors independently derive a new signature from cohort 2a?

Text corrections:

  1. In the abstract, it is not clear that “poor prognostic lipid profile” refers to a specific 3-lipid signature.
  2. Figure 2 legend needs to specify the exact composition of 3LS.
  3. The 3LS algorithm should be provided in the methods to allow independent replication of the study. What is meant by “aligning the lipidomic dataset”?
  4. Line 215 Figure 2b should be 2a.
  5. Line 219 Figure 2c should be 2b.
  6. Figure 3 headings for panel b and panel c, “3Ls” should be “3LS”
  7. Figure 5 heatmaps would be best replaced with summary visualisation. The complex heatmaps may be provided as supporting information.
  8. Is IL18 in Line 339 meant to be IL8?

Reviewer 2 Report

In the manuscript, the authors show the relationship between circulating lipids and cytokines in metastatic castration-resistant prostate cancer through cytokines and lipidomic analysis toward patient plasma samples. They found higher baseline levels of sphingolipids including ceramides were associated with shorter overall survival. it suggests that circulating lipids can be used as prognostic biomarkers. The whole flow was organized well and data was presented in detail. However, there are some issues that need to be improved.

  1. the authors need to provide background or discuss the effect of castration therapy on the body inner environment;
  2. the relationship between cancer metastasis and circulating lipids/cytokines need to be shown;
  3. what is the relationship between circulating lipids/cytokines and tumor environment?

Round 2

Reviewer 1 Report

Thank you for the changes, most of which were satisfactory.  

However, I disagree on the utility and readability of heatmaps. Not just Figure 5, but also Figure 4 - the cytokine/lipid correlations are buried in the complex heatmap shown. I highly recommend the authors to improve visualisation so the key findings are clear to the readers. 

The new Figure S4 is informative, but as the correlation value is 0.29 or 0.3, it should be stated as "weakly" correlated. It is significant but weak.

Furthermore, the prognostic changes in lipids and cytokines (Figure 6) are also divergent. On reading the manuscript, it seems the authors are forcing the interpretation of a link between ceramide and cytokine via S1P which is not directly supported by the presented data. Although other papers have reported such data, it is not proven in the current manuscript, hence it would be desirable to reduce the narrative of direct mechanism but rather focus on better presenting the associations identified. To this end, a summary figure or proposed biomarkers/mechanisms will help the reader to clarify the multiomics data reported in this paper.  

Text corrections. 

  1. In Simple Summary, "the levels of one lipid (ceramide)" - clarify if this is one lipid class, or one lipid species. If the latter, specify the exact ceramide.
  2. New Figure S4 title should be 'prognostic lipids' not 'cytokine', no cytokines shown in this figure.

Round 3

Reviewer 1 Report

Thank you for the revisions. The paper is now ready for publication with the following minor corrections.

Section 3.5, references to Pearson coefficients and p-values should be referred to Figure 4b, or a supplementary figure as appropriate. Figure 4a does not show any correlations so referring to "Figure 4" is not correct.  

Line 451-457 should not be bolded.

Line 457-458, Section 5 should start on a new line. 
